# Does the Compact City Paradigm Help Reduce Poverty? Evidence from China

**DOI:** 10.3390/ijerph19106184

**Published:** 2022-05-19

**Authors:** Lu Liu, Yu Tian

**Affiliations:** School of Business, Sun Yat-sen University, Guangzhou 510275, China; liulu68@mail2.sysu.edu.cn

**Keywords:** compact city, urban poverty, sustainability, urban shape

## Abstract

City shape is an essential reflection of spatial structure, but it has largely been ignored in urban form research. This study employs night-time satellite imagery to depict the scope of urban economic activity to investigate its impact on urban poverty. It is the first study to provide a comprehensive assessment of the mechanisms of city shape on urban poverty by using the fixed-effect estimate methodology for panel data of 285 Chinese cities from 2000 to 2018. The results showed that city compactness has an inverted U-shaped relationship with poverty incidence, which was verified by several robustness tests. Compactness can significantly attract more population into the city, and space costs and commuting costs are important influence channels. Furthermore, there exists heterogeneous nexus between city shape and urban poverty. Compactness has more significant poverty reduction effects in low-attractive cities with low productivity, low wages, and high illiteracy rates.

## 1. Introduction

The concept of sustainable development is development that can be sustained in the long term without compromising the ability of future generations to meet their own needs [1,2]. The compact city is a conceptual model for sustainable urban development in response to urban sprawl [3,4,5], emphasizing the use of mixed land resources and intensive development strategies to improve the efficiency of urban land use and the quality of urban development. Poverty eradication is the primary goal of sustainable development. However, little is known about whether compact cities can contribute to poverty reduction.

Over the last two decades (year 2000–2020), the global urban population has increased from 46% to 56% of the total population. According to the United Nations, this proportion is expected to reach nearly 68% by 2050. As urbanization accelerates in many countries, urban issues will become increasingly important to socio-economic development, thus improving awareness and understanding of poverty in an urban context will become increasingly important [2,6].

The compact city has been proven to play a role in promoting sustainable development, such as improved productivity [7], reduced pollution [8,9,10,11,12,13] and for having smaller ecological footprints [14] and better city health [15,16]. However, compact cities have also had some negative effects, such as a lack of urban green space, overcrowding, and social withdrawal [9,11,12,17,18]. We discovered that these studies focus mainly on the impact of compactness on the quality of life of the group as a whole, ignoring the impact on specific groups, particularly the urban poor. Burton [9] found that compact city (higher urban densities) may be positive for some aspects of social equity and negative for others. This serves as a reminder that the effects of city compactness can be highly heterogeneous, with different effects on different groups in society. Understanding whether compact cities can truly improve the lives of the urban poor can assist policymakers in making decisions about urban planning and city compactness.

China, which has experienced rapid urbanization, provides an excellent context for understanding the issue of urban poverty. Using panel data of 285 cities from 2000 to 2018, we applied a fixed-effect estimation method to empirically evaluate the relationship between city compactness and urban poverty. We assessed whether compactness could affect poverty levels, exploring which possible channels and to what extent urban poverty is affected by compactness. Furthermore, we investigated the heterogeneous nexus between city compactness and urban poverty from a variety of perspectives (i.e., the level of productivity, the level of wages, and the educational level of residents).

The main contributions of our study can be summarized in three aspects. (1) It provides a new perspective for research on urban poverty reduction. To the best of our knowledge, no previous study has investigated the effect of city compactness on urban poverty with long-panel data. (2) This study investigates how urban compactness affects urban poverty, including the indirect effects of city shape attributes. The results would uncover latent causal mechanisms and offer a new perspective for further relevant research. (3) This study adds to the empirical evidence on the impact of compact cities, particularly in high-density cities. The debate between supporters and detractors of compact cities over urban morphology has become polarized, partly due to the over-emphasis of the American experience while ignoring other urban backgrounds [19]. In contrast to American cities with low density and mature urbanization, Chinese cities can represent high-density and rapid urbanization areas.

The remainder of our paper is organized as follows. Section 2 proposes the research hypotheses. Section 3 presents econometric specifications and data. Section 4 reports the empirical results, covering the direct impact of city shape on urban poverty, heterogenous tests, mechanism tests, and some robustness tests. Section 5 provides the discussion of results and the conclusions.

## 2. How Does City Shape Affect Urban Poverty?

There has been no definitive consensus on how and how much city compactness affects urban poverty. In this section, we discuss possible explanations detailing the relationship between city compactness and urban poverty.

Headcount ratio, the proportion of the poor to the total population, is the most basic and widely used method for assessing the degree of poverty [20,21]. In this article, when we talk about poor people, we are referring to the “lower-income” class. Two main reasons explain why a city’s poverty level declines. The first is that the living standards are improved, lifting people out of poverty. The second is that poor people move into other cities for various reasons (e.g., the rising cost of living). The following subsection provides a detailed analysis of the potential impact of compact cities on total population and poor people.

### 2.1. Population Growth

The shape of a city is widely considered to affect location selection. When families or businesses decide where to relocate, they may consider city shape when weighing the benefits and drawbacks of various locations. Why is city shape important in location selection? We will address it separately from the standpoints of firms and families.

For firms, productivity is a major consideration in decision-making. When companies make investment decisions, they mainly consider differences in inter-sectoral and inter-regional productivity over other factors. The compactness and shape of a city can have various effects on productivity [22]. Compact markets may allow for a more efficient sharing of indivisible facilities (e.g., local infrastructure), risks, and gains from variety and specialization. They can also facilitate better matchmaking between employers and employees, buyers and suppliers, joint projects partners, and entrepreneurs and financiers. In addition, compact markets can help facilitate and expedite the learning of new technologies, market evolutions, and new forms of organization. Due to these benefits, companies may be more likely to prefer highly compact areas.

Compact cities can also provide a number of benefits to families, such as better service delivery and increased accessibility since establishments and facilities are closer to one another. If a city’s compactness could result in greater efficiency, it would significantly increase the city’s appeal. Additionally, several studies have found that compared with low-density suburban residents, compact city residents feel that their needs are addressed to a greater degree [15] and that they achieved significantly higher social well-being [23]. 

If compact cities do indeed provide these benefits, then they are more likely to have larger populations. This would then suggest that city shape significantly influences the location choices of households and businesses. Based on these arguments, we propose the following hypothesis:

**Hypothesis 1 (H1)**.
*Urban compactness is positively correlated with population growth. As cities become more compact, the population grows faster.*


After analyzing the impact of a city’s compactness on the total population, we explore its effect on poverty levels. Food, clothes, housing, and transportation are the four basic requirements of people. Among them, housing and transportation, i.e., space costs and commuting costs, have the most significant influence on the quality of life and location choices, particularly among low-income families. Therefore, we focus the analysis on these two aspects.

### 2.2. The Housing Impact for the Poor

The cost of housing is a crucial factor determining disposable income and life quality for low-income households. Extreme lack of affordability can contribute to homelessness for the least advantaged. In China, despite the government’s efforts of building low-rent and affordable housing in various locations for poor households, housing prices began to rise rapidly at the beginning of the 21st century, putting heavy burden on the urban poor. By understanding how city compactness affects residential affordability, decision-makers would be able to adopt changes and respond more effectively.

As urban areas become more compact, urban population growth tends to accelerate. A large influx of population naturally leads to an increase in demand for urban housing. Similarly, urban morphology influences housing supply. As compactness improves, urban land becomes scarcer and more expensive [24,25,26], resulting in a major reduction in the housing supply. Figure 1 shows the impact of city compactness on the supply and demand for housing. In general, for cities with low compactness, the demand for housing is less than the supply. This causes housing prices to decline, and a large number of houses are vacant. In contrast, for highly compact cities, housing demand exceeds supply, leading to rising housing prices. As a result, with the increase in urban compactness, housing will turn from surplus to scarcity, and housing prices will turn from decline to rise. Thus, we hypothesize the following:

**Hypothesis** **2.1** **(H2.1).**
*There is a nonlinear relationship between city shape and housing prices. When compactness is low, increasing compactness helps to lower housing prices. When the compactness is high, however, housing prices rise as the compactness improves.*


It is a widely held view that urban places are not neutral spaces in terms of their impact on people’s lives [9,27]. In this study, the impact of housing prices is also assumed to be non-neutral for the urban poor. On the one hand, high house price is a pull factor for the poor’s location choice. This is because high house prices frequently indicate more employment opportunities and amenities, whereas low house prices indicate macro-structural weaknesses in an area or reduced employment opportunities. According to this viewpoint, cities with high house prices are appealing to the poor, while low-housing price cities are not preferred by the poor. As a result, areas with high housing prices tend to attract more poor people. On the other hand, high house price is also a push factor for the poor’s location choice. According to Roback [28], house prices are the primary cost of living and working in cities, and they have a direct influence on the decision to live there. Low-cost housing can be attractive to migrants, particularly for impoverished residents, while high housing prices deter migration can serve as entry barriers for the poor, resulting in a negative relationship between housing price and poverty headcount ratio [29]. When deciding on a location, it is critical to fully consider the costs (rents) and benefits (income) of regional housing prices. The impact of high house prices on the poor is two-fold: it increases the poor’s living burden while also bringing economic opportunities, implying that the poor must weigh the cost of living against employment opportunities. When the push factor outweighs the pull factor, house prices are negatively correlated with the incidence of poverty; when the pull factor outweighs the push factor, the opposite is true.

We should consider not only the impact of house prices on poor people’s location choices, but also the impact of house prices on the quality of life of local residents. Rising house prices have added to the urban population’s burden, potentially pushing more people into poverty. High house prices may lead to a decline in the quality of life of local residents and an increase in urban poverty if the increase in the cost of living caused by high house prices is greater than the economic benefits brought by job opportunities. On the contrary, high house prices may improve the quality of life of local residents and reduce urban poverty.

Thus, we hypothesize the following:

**Hypothesis** **2.2a** **(H2.2a).***Housing price is negatively related to the incidence of poverty*.

**Hypothesis** **2.2b** **(H2.2b).***Housing price is positively related to the incidence of poverty*.

### 2.3. The Transportation Impact for the Poor

The urban poor face enormous challenges in their daily lives. Many live in densely populated city spaces or in more remote suburban areas with limited access to jobs and social services. Inconvenient or prohibitively expensive transportation costs may significantly hamper people’s mobility and access to essential facilities and services. This may result in low living standards and even increased poverty levels. Transportation is crucial to the urban poor, and understanding the impact of traffic accessibility on impoverished communities is vital in developing poverty reduction policies and improving people’s living standards.

First, we need to understand the basic situation in urban transportation. For short intra-city travel, walking, bicycling, buses, taxis, and private cars are popular modes of transportation. However, in recent decades, urban transportation has increasingly been dominated by buses and private automobiles. For many travelers who prioritize accessibility and speed, public transportation is not often their first choice. Instead, private cars have become their preferred mode of transportation. In China, there were about 226.35 million private vehicles in 2019, 309 times the number in 1989, 42 times the number in 1999, and 5 times the number in 2009. However, for those with less income, they rely heavily on low-cost modes of transportation such as walking, bicycles, electric vehicles, or buses. Thus, there are significant differences in travel between the poor and the non-poor that should be considered when analyzing the impact of compact cities.

It is generally believed that compact cities have the potential to reduce the distance between home and work and reduce the use of private cars, saving time and money spent on commuting [30], and public transportation is considered to work better [31]. In this study, we posit that the relationship between urban compactness and transportation is not necessarily linear. At different levels of compactness, the improvements in transportation infrastructure can vary considerably. When urban compactness is low, demand for motor transportation rises rapidly, so that the rate of vehicle ownership growth is often faster than the rate of public transport to accommodate traffic growth [32]. Piecemeal transportation development to alleviate traffic bottlenecks on the road network frequently comes at the expense of the poor. Private cars reduce the passenger flow of public transport and eliminate the financial feasibility of the public transport system, resulting in the decline of service quality and quantity. In cities with low compactness, the growth trend of private vehicle use plays a leading role in terms of convenience.

When urban compactness is relatively high, infrastructure improvements would include projects catering to high population density, such as buses and subways. This is because compact cities can improve public transportation utilization by strengthening land use near public transportation stations, which greatly improves the feasibility of public transport supply [33]. Newman and Kenworthy [34] found indirect evidence that higher-density cities are associated with high use of public transport. As public transport declines, population density drops at around 20–30 people per hectare [35]. As cities become more compact, commuting distances decrease, the benefits of public transportation infrastructure also become more apparent, thus the demand for private cars decreases. Based on the discussion, we propose the following hypothesis:

**Hypothesis** **3.1** **(H3.1).***The relationship between city shape and private cars is inverted-U-shaped*.

**Hypothesis** **3.2** **(H3.2).***The relationship between city shape and public transportation is U-shaped*.

For the two modes of transportation, the impact on the poor must be discussed separately. The expansion of public transportation will attract the poor, whereas the increase of private cars will reduce the poor’s quality of life. In less compact cities, public transportation is relatively poor. This would then cause the number of private cars to grow, but low-income families are unable to afford them. They would then have to deal with long-distance commutes and heavy traffic. The difficulties and high costs associated with long commutes may also result in missed job opportunities, further hindering economic mobility. In comparison, infrastructure in densely populated cities is often better, providing better access to public transportation. As cities become more compact, the number of private cars decreases, traffic congestion decreases, and the difficulty and cost of commuting for the poor decreases. According to Glaeser et al. [36], public transportation is an important policy instrument that can influence the location decisions. The location preference of low-income families is highly sensitive to traffic conditions. When commuting conditions are poor, low-income families tend to emigrate, thus reducing the number of poor people in the city. In contrast, favorable commuting conditions and good infrastructure may attract low-income migrants, causing urban poverty levels to rise. Therefore, we expect the following:

**Hypothesis** **3.3** **(H3.3).***The number of private cars is positively related to urban poverty*.

**Hypothesis** **3.4** **(H3.4).***Public transportation is positively related to urban poverty*.

Urban compactness indicates the state of compactness of a given city. When the city’s compactness is low, we assume it is not compact and may even spread disorderly. When the city’s compactness is high, the distribution of urban economic activity is also compact. Based on the above discussion on total population growth and the impact of housing and transportation on the poor, we argue that the impact of compact cities on urban poverty is nonlinear and complex. We therefore hypothesize:

**Hypothesis** **4** **(H4).***The impact of city shape on urban poverty is nonlinear*.

## 3. Econometric Specifications and Data

### 3.1. Model Specification and Empirical Methodology

Consistent with existing literature exploring the determinants of urban poverty [37,38,39], this paper empirically examines the impact of city compactness on urban poverty by estimating the following two equations.
(1)$$PHR_{i,t}=\alpha *nCohesion10_{i,t}+\gamma_{1}*C_{i,t}+\delta_{i}+\epsilon_{t}+\varepsilon_{i,t}$$
(2)$$PHR_{i,t}=\beta_{1}*nCohesion10_{i,t}+\beta_{2}*nCohesion10sq_{i,t}+\gamma_{2}*C_{i,t}+\delta_{i}+\epsilon_{t}+\varepsilon_{i,t}$$
where $PHR_{i,t}$ is the incidence of urban poverty, $nCohesion10_{i,t}$ is the normalized cohesion index, $nCohesion10sq_{i,t}$ is the square of the normalized cohesion index, and $\delta_{i}$, $\epsilon_{t}$ are the city fixed effect and year fixed effect. $C_{i,t}$ consists of a series of control variables, such as GDP per capita (*lnpcgdp*, deflated by CPI and in log), the ratio of fiscal expenditure to GDP (*fiscal*), total export–import volume (*lntrade*, deflated by CPI), average years of education (*lnays*), and industrial structure (*IndStr*, the ratio of tertiary sector added value to secondary sector added value). Note that *i* (*i =* 1…285 cities) and *t* (*t =* 2000…2018 years) denote city and year, respectively.

In addition to the variables included in the model, there are many characteristics that are unique to each city, such as social customs, history and culture, all of which influence urban poverty. These factors are difficult to characterize but do not change over time, so the fixed effects model is an effective method for addressing such problem.

### 3.2. Variables, Data Source, and Summary Statistics

Two datasets were used to investigate the effect of city shape on urban poverty: the Night-time Lights dataset and Urban Minimum Living Standard Security dataset.

First, we used the DMSP/OLS Night-time Lights dataset (Recorded by the Operational Linescan System (OLS) from the US Air Force Defense Meteorological Satellite Program (DMSP)) from 2000 to 2013, and the NPP/VIIRS Night-time Lights dataset (Recorded by the Visible Infrared Imaging Radiometer Suite (VIIRS) on the Suomi NPP satellite) from 2014 to 2018. Each year, the dataset contains nighttime satellite images that record the intensity of Earth-based lights. We used imagery to delineate urban areas by taking into account spatially contiguous lighted pixels surrounding a city’s coordinates that have a luminosity greater than a predefined threshold of 10. The shape compactness of the urban footprints was then quantified for each year. The degree to which a polygon’s shape deviates from a circle is defined as shape compactness. A variety of indexes can be used to measure shape compactness; for this study, we used the cohesion index, an indicator commonly used in urban planning [40,41]. 

Cohesion is a natural indicator of a city’s overall accessibility. The proximity of objects to one another is the focus of the cohesion property of geographic shapes. The more cohesive a metropolitan area is, the more accessible it is to its residents. The average Euclidean distance (in kilometers) between two points within a polygon is used to calculate the cohesion index. Larger distances between points and less compact shapes are associated with higher cohesion index values. It should be noted that any compactness index based on distance within a polygon is mechanically correlated with its area. The footprint area was controlled in the cohesion index computation to separate the geometry effect from the city size. It is also worth noting that the normalized cohesion index, *nCohesion10*, is used in this paper.

We then used the dataset containing the number of people receiving minimum living allowances from China’s provincial Civil Affairs Bureau. This dataset includes statistics from the Minimum Living Standard Guarantee (Dibao) Program, which providing cash assistance to families whose income per capita is less than the minimum living standard set by the local government. The recipient of the security fund must have a permanent non-agricultural household registration in this city and cannot own high-end consumer goods such as cars and motorcycles that are not required for basic living. We used this dataset to calculate the poverty headcount ratio. The headcount ratio is the proportion of those living below the poverty line to the total population; it is the most basic and widely used method for assessing the degree of poverty [20,21]. The poverty headcount ratio (PHR) is given by the expression:(3)PHR=Q/P
where P represents the total population, and Q represents the number of poor people. These data were also used to calculate the location quotient (*LQ*) of poverty for the robustness test. The calculation formula of *LQ* is described in detail in the section of robustness test.

To have a general understanding of the levels and changes of urban compactness and the incidence of poverty, we drew line charts for the time-series data (see Figure 2). In general, urban compactness was found to be increasing while the incidence of poverty was decreasing. As shown in the figure, there was a negative correlation between the two variables.

To account for the effects of extraneous variables, we included five control variables: *lnpcgdp*, *fiscal*, *lntrade*, *lnays*, and *IndStr*. These data were obtained from China City Statistical Yearbook, China Statistical Yearbook For Regional Economy, and statistical yearbooks at the provincial and city levels. Due to some missing values, we also referred to other data sources, such as the Wind Economic Database and the CEIC (https://insights.ceicdata.com/node/CN, accessed on 17 May 2022).

Finally, to check the possible impact channels through which city shape affects urban poverty, we used three other variables. To examine whether city shape can affect urban poverty through the total population growth rate, we employed the growth rate of the permanent population (*PopGR*) parameter. We also created the real housing price (*HsPr_real*) parameter to determine whether city shape can affect urban poverty through house burden. To test whether city shape can affect urban poverty through private cars, we constructed the number of private car ownership (*lncarpc*) parameter. The data sources are consistent with the control variables. The descriptive statistics of variables are shown in Table 1.

## 4. Empirical Results

### 4.1. The Impact of Compactness on Urban Poverty

Table 2 presents the results of the Panel Fixed-effects model on the effect of city shape on urban poverty. The results show that the coefficient for *nCohesion10* in Column (1) is not significant. The coefficients for *nCohesion10* and *nCohesion10sq* in Column (2) are 0.887 and −0.492, respectively, significant at the 1% level. These results indicate that the relationship between city shape and poverty is not linear but an inverted U-shaped. At the right side of the axis of symmetry, the increase in urban compactness reduces the incidence of poverty. Hence, Hypothesis 4 is confirmed. 

To check whether the inverted U-shape occurs within the range of the data, we plotted the symmetry axis of the inverted U-shaped curve. As shown in Figure 3, the location of the symmetry axis is approximately 0.90. The figure shows a left-skewed distribution (also called negatively skewed distribution). Combined with the centile estimates for *nCohesion10* (see Table A1), about 74% of the cities have shape compactness greater than 0.90. This means that when the urban compactness is lower than 0.90, improvements from urban compactness do not support poverty reduction; when the urban compactness is higher than 0.90, the improvements help alleviate poverty (Table A2 in the Appendix A details the ten most and the ten least compact cities in the dataset). 

For the control variables, although some of the coefficients were not statistically significant, some reasonable conclusions can still be drawn. The results suggest that achieving high-income growth does not ensure poverty reduction at a similar pace and that improvements in educational outcomes are strongly and negatively associated with poverty incidence. These support the findings of Awan et al. [42] and Janjua and Kamal [43]. 

### 4.2. Heterogeneous Tests 

We then analyzed the relationship between city shape and urban poverty, based on productivity, illiteracy rate, and salary. The purpose of this section was to explore whether the effect of city shape on urban poverty will change over the different city environments. We used a sub-sample regression method to divide the total sample into low-productivity and high-productivity groups, low-illiteracy and high-illiteracy rate groups, and low-salary and high-salary groups. Cities with productivity lower than the sample median were classified as low-productivity, while those with productivity higher than the sample median were grouped into high-productivity. The illiteracy rate and wages were classified similarly. We then re-estimated our basic model. Table 3 shows the results for the different samples. Columns (1) and (2) display the estimation results for the high-productivity and low-productivity groups, columns (3) and (4) for the high-illiteracy and low-illiteracy rate groups, and columns (5) and (6) for the high-salary and low-salary groups. 

When the results are compared across different groups, a significant inverted U-shaped correlation was found between city shape and the poverty incidence in cities with low productivity, high illiteracy, and low salary. In contrast, the effect of city shape is not significant in other scenarios. This means that becoming more compact in cities with high productivity, high wages, and low illiteracy does not produce poverty reduction effects.

### 4.3. Mechanism Tests

We then analyzed which channels city shape affects poverty. Specifically, we examined whether the city shape affects low-income residents through the population, housing, and transport effects. In this analysis, we replaced the explained variable *PHR* with *PopGR*, *HsPr_real*, and *lncarpc* and *bus*. Table 4, Table 5 and Table 6 show the results of the mechanical tests for the four explained variables *PopGR*, *HsPr_real*, and *lncarpc* and *bus*. 

In Table 4, the coefficient of *nCohesion10* is positive and significant in Column (1), indicating that compact cities attract more people. The coefficient of *nCohesion10sq* is not significant in Column (2). This means that the influence of city shape on population growth is linearly positive, which is consistent with the conclusions of Harari [41]. We can confirm Hypothesis 1.

Table 5 summarizes the estimated results for the housing price effect. We found the effect of city shape is not significant on real housing prices in Column (1), while significant in Column (2). The coefficient for *nCohesion10* is −50,082.4, and 28,200.4 for *nCohesion10sq*. There is a U-shaped relationship between city shape and housing price, indicating the housing price effect for the correlation between city shape and urban poverty. We can confirm Hypothesis 2.1. Column (3) presents the coefficient of housing prices on urban poverty. The impact is not significant. To check whether the mediating effect is significant, we performed the Sobel test. The estimated results for the left and right sides of the symmetry axis (*nCohesion10* = 0.90) are shown in columns (4) and (5), respectively. The results show that the mediation effect is negative and significant, indicating that high house prices have a greater crowding out effect than income effects, squeezing out the poor and lowering the quality of life for the local low-income population. Foote [44] also finds that for a homeless workforce, high housing costs increase the likelihood of leaving, which is consistent with the findings of this paper. We can confirm Hypothesis 2.2a.

In Table 6, the coefficient of *nCohesion10* is 0.967 in Column (1); the linear effect is not significant. In Column (2), the coefficients of *nCohesion10sq* is −12.47, also statistically insignificant. The results suggest there is no significant relationship between city shape and private cars. Most studies suggest that compact cities offer the possibility of reducing car use [15]. In this paper, however, we find no evidence that compact urban layout reduces car use. Similarly, in Column (4), the coefficient of *nCohesion10* is not significant. In Column (5), the coefficients for *nCohesion10* and *nCohesion10sq* are −3.911 and 2.293, statistically significant at 5% level, indicating a U-shaped relationship between city shape and public transport. We can confirm Hypothesis 3.2. In Column (3), the coefficient is significant and positive, confirming Hypothesis 3.3; and in Column (6), the coefficient is significant and positive, confirming Hypothesis 3.4. While bus development benefits the poor, it is also positively related to urban poverty. This does not mean that public transportation is bad for the poverty. Instead, low-income residents are often drawn to good public transportation, which raises the city’s poverty rate. This conclusion is consistent with the findings of Glaeser et al. [36].

### 4.4. Some Robustness Tests 

To check the robustness of our results and obtain more accurate conclusions, we conducted several robustness tests. The first robustness test was to refine the sample. Some studies on regional disparities suggest that there is a statistical difference when municipalities are included and excluded from the samples. We deleted the four municipalities directly under the central government from the samples. As shown in Table 7, the linear impact of city shape on poverty was still not significant, while the inverted U-shaped relationship was significant. The municipality factor did not significantly change the results, and the robustness test supports the earlier conclusions. 

The second robustness test made use of alternative poverty indicators. We used the location quotient as the dependent variable. The location quotient is an index used to measure the spatial distribution of elements in an area that reflects a certain degree of sector concentration. It is the ratio between the factor percentage in a subregion and the factor percentage for the entire area. The location quotient (*LQ*) is one of the most basic analytical tools in economic development research [45], which has since been widely used in different fields, such as industrial concentration [46]. Here, we used the same method to measure the concentration of the population living in poverty. The formula for the location quotient (LQ) is given by the expression:(4)$$LQ_{ij}=\left(Q_{i}/\overset{m}{\underset{j=1}{\sum }}Q_{i}\right)/\left(P_{i}/\overset{m}{\underset{j=1}{\sum }}P_{i}\right)$$
where $Q_{i}$ is the number of poor population in city i; $P_{i}$ is the total number of population in city i; and m is the number of cities. If LQ > 1, the concentration level of the poor population in a particular city is higher than the national average level. The greater the value of LQ, the higher the concentration level of the poor population.

Columns (1)–(2) in Table 8 present the estimation results. The coefficient for *nCohesion10* is insignificant in Column (1), while the *nCohesion10* coefficient is positive, and the *nCohesion10sq* coefficient is negative, both significant at the 1% level. The robustness test results are in line with our baseline results.

We performed additional robustness tests. First, we added a cubic term to the model to determine whether the effect of city shape on poverty is an inverted U-shape or S-shape. In Column (1) of Table 9, the coefficient of the cubic term is not significant, supporting our U-shaped relationship hypothesis. Additionally, we controlled for the interaction between region and time based on the baseline regression. As shown in Column (2) of Table 9, the basic conclusions are still robust. 

## 5. Discussion and Conclusions

In the past two decades, the increasing trend of urban poverty has largely threatened and impeded sustainable development. Understanding the external factors causing the increase in urban poverty would help promote sustainable urban development and improve the well-being of impoverished communities. In this study, we explored how the geographic changes in urban economic activities influence poverty levels. Notwithstanding data and technical limitations, this article contributes to the debates on the compact city. 

The empirical results suggest that the compactness of urban economic activities has an inverted U-shape relationship with the incidence of poverty. This finding is confirmed by several robustness checks, including sample refinement and changing the explained variable. Specifically, further developments in compact cities are conducive to poverty alleviation and promote sustainable urban development. However, this conclusion has certain preconditions. First, the city should have a high degree of compactness (greater than 0.90, more than 74% of the cities in China meet this condition). Second, only in cities with low productivity, high illiteracy rates or low wages can compactness have the effect of poverty reduction. In cities with high productivity and high wages, the increase of urban compactness has attracted more poor people. These results indirectly reflect the importance of high productivity and high wages for the urban poor. It is difficult to improve life in cities with low wages (low productivity), but cities with high wages (high productivity) make it possible for the poor to obtain income. Therefore, the poor are willing to move to developed cities.

Possible causal mechanisms that explain the main effect were also explored. The results found that compact cities are neither completely harmful nor beneficial to vulnerable urban groups. Compact cities can attract highly competitive enterprises, bringing more employment opportunities and attracting more population inflows, which is consistent with the conclusions of Harari [41]. Increased public transportation also attracts more poor people into cities. This is consistent with the prevailing policy view that cities need to provide more public transportation and access to it for low-income and immigrant populations [47]. However, we found that high housing prices are bad for the poor. Although high housing prices reduce poverty, this is because high housing prices create crowding out of low-income populations. Increased housing prices in compact cities may increase the burden for low-income immigrants, which means that they may lose suitable employment opportunities due to high housing prices. Good public transportation can attract more poor people, but because of the cost of high housing prices, low-income populations are more likely to be crowded out. In addition, the rapid growth in private car use may also be particularly harmful to low-income residents.

Although the overall trend is that urban development is becoming more and more compact, there are also some cities experiencing shape deterioration. The non-compact distribution of urban economic activities means longer commuting distances for the city’s residents. To prevent the deterioration of connectivity caused by rapid development and reduce the negative impact of bad form, the government needs to reduce barriers to mobility between cities to ensure the growth of urban population, so as to bring poverty reduction effect through agglomeration. Secondly, controlling the number of private cars and improving infrastructures, such as buses and subways, will help shorten the commuting time within cities, which would improve the welfare of the urban poor.

There are also some limitations to this paper. Firstly, poverty incidence calculated using the number of people receiving subsistence allowance is an excellent indicator for overall poverty. The use of the poverty location quotient as a proxy poverty indicator also enhances the credibility of the results. However, the use of such a concept still has certain limitations. Specifically, it does not reflect the multidimensional poverty problems, such as the health and education problems of the poor, which are increasingly prominent in cities. More complex conceptual models and methods are needed in future research to address these limitations. Secondly, this study did not differentiate between variations in regional forms (e.g., cluster distribution, stellate distribution, ribbon distribution). In future research, it would be interesting to provide more theoretical micro-foundations for the influences of more dimensions of urban form on the disadvantaged in the urban areas and clarify how urban form plays a critical role. Finally, note that there are considerable differences in development patterns, especially between developed and developing countries. Subsequent studies can further explore these differences and compare our results with those conducted in other areas with different socio-economic conditions.

## Figures and Tables

**Figure 1 ijerph-19-06184-f001:**
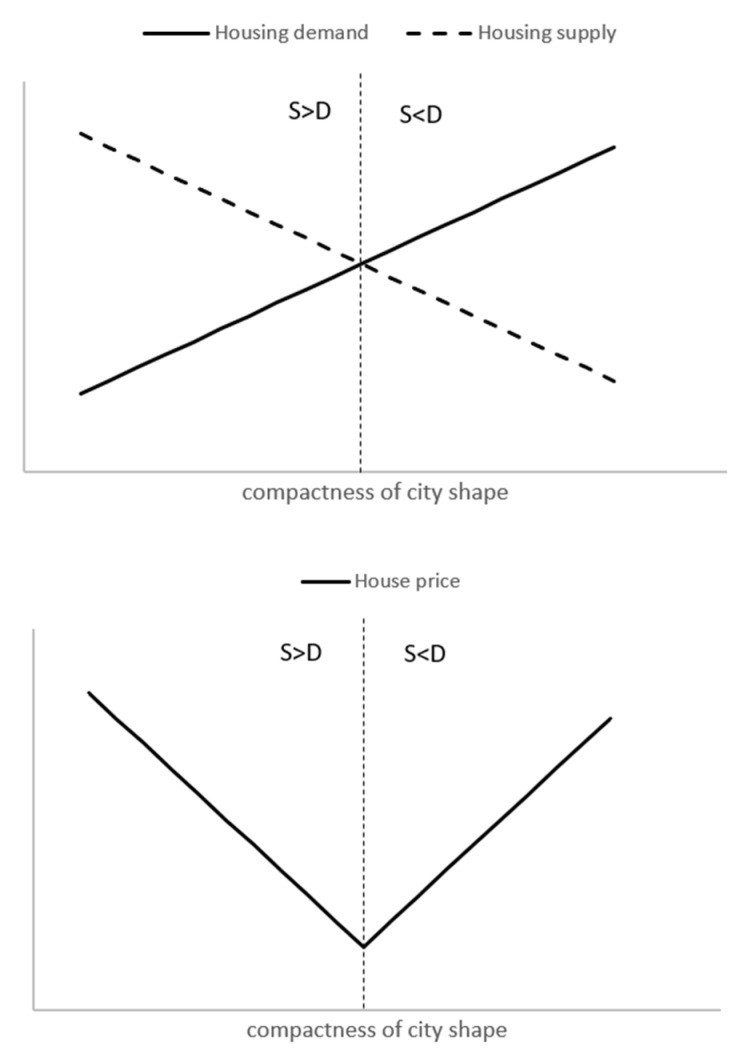
How does the compactness of the city shape affect housing prices? Notes: S < D stands for “supply falls short of demand”, and S > D stands for the inverse.

**Figure 2 ijerph-19-06184-f002:**
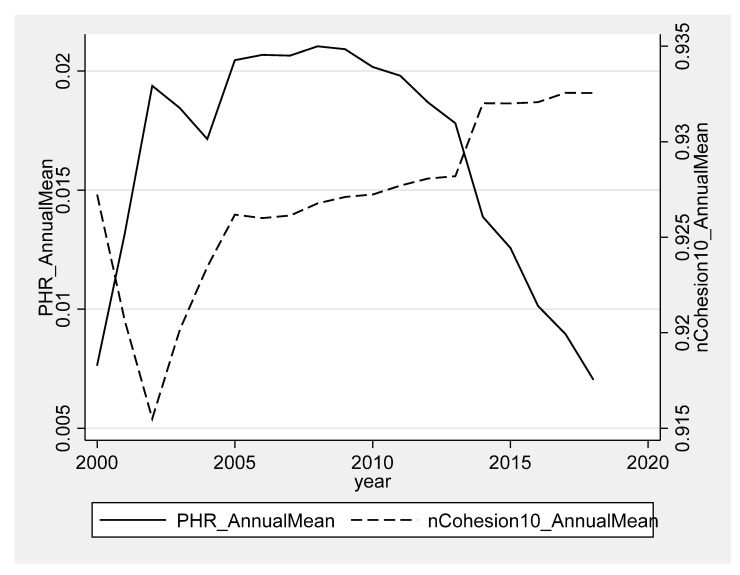
Trends in urban shape and poverty from 2000 to 2018. Notes: The left vertical axis is the annual mean of *PHR* and the right is the annual mean of *nCohesion10*. On the horizontal axis is the year.

**Figure 3 ijerph-19-06184-f003:**
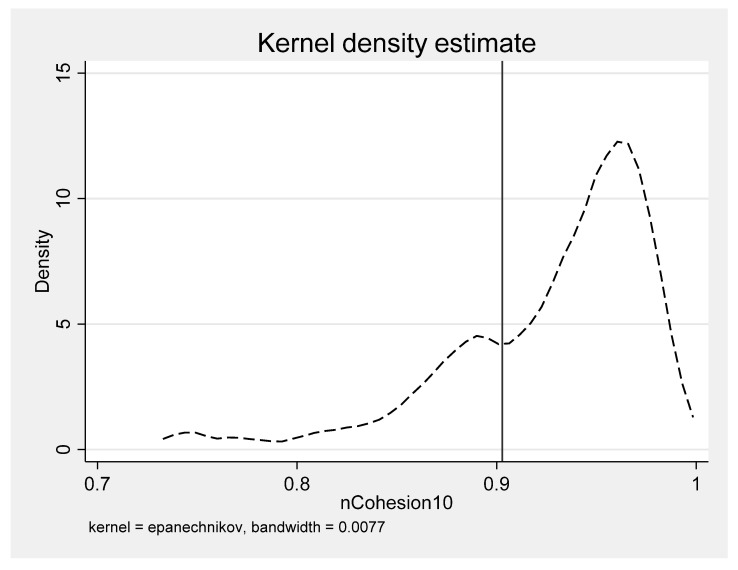
The Epanechnikov kernel density estimate of *nCohesion10.* Notes: The solid line acts as the symmetry axis of the inverted-U shape.

**Table 1 ijerph-19-06184-t001:** Descriptive statistics.

Variable	Obs	Mean	Std. Dev.	Min	Max
*PHR*	4145	0.0169	0.0156	0.000727	0.0843
*LQ*	4145	1.292	1.134	0.0664	5.935
*PopGR*, %	5348	0.607	3.566	−13.43	18.35
*HsPr real*, CNY/m^2^	5364	2676	1852	680.3	11,763
*car_pc*, vehicles/10,000 people	3848	687.2	740.0	22.34	3937
*bus*, 10,000 vehicles	5308	1166	2139	46	14,852
*TFP*	5364	1.506	0.760	0.103	2.898
*IlliteracyRate*, %	5364	7.563	3.623	2.220	18.63
*lnSalary*, CNY, in log	5364	9.944	0.578	8.721	11.01
*nCohesion10*	5364	0.927	0.0514	0.741	0.991
*lnpcgdp*, one thousand CNY, in log	5364	2.871	0.814	1.053	4.598
*fiscal*	5364	0.143	0.0880	0.0119	0.484
*lntrade*, one billion CNY, in log	5364	3.753	2.192	−1.512	9.380
*lnays*, years, in log	5364	2.040	0.0652	1.935	2.247
*IndStr*	5364	0.890	0.417	0.223	2.629

Notes: Std. Dev. represents standard deviation.

**Table 2 ijerph-19-06184-t002:** Impact of city shape on urban poverty.

	(1)	(2)
Variable	BM1_0	BM1_1
*nCohesion10*	0.0252	0.887 ***
	(0.79)	(2.67)
*nCohesion10sq*		−0.492 ***
		(−2.64)
*lnpcgdp*	0.00351 ***	0.00379 ***
	(2.77)	(2.98)
*fiscal*	0.0168 ***	0.0161 ***
	(3.63)	(3.50)
*lntrade*	−0.0000884	−0.0000358
	(−0.26)	(−0.11)
*lnays*	−0.0295 ***	−0.0297 ***
	(−3.61)	(−3.66)
*IndStr*	−0.00133	−0.00134
	(−1.06)	(−1.06)
*_cons*	0.0355	−0.339 **
	(1.13)	(−2.34)
r2_a	0.3198	0.3275
F	20.7762 ***	19.9923 ***
N	4145	4145

Notes: *t* statistics are in parentheses. **, and *** represents *p* < 0.05, and *p* < 0.01, respectively. For concise presentation, the coefficients for the year dummies are excluded from the tables.

**Table 3 ijerph-19-06184-t003:** Heterogeneous effects: impact of city shape on urban poverty.

	(1)	(2)	(3)	(4)	(5)	(6)
	TFP_h	TFP_l	IlliteracyRate_h	IlliteracyRate_l	Salary_h	Salary_l
*nCohesion10*	0.462	1.440 ***	0.952 **	−0.178	0.386	0.441 **
	(1.47)	(2.73)	(2.38)	(−0.32)	(1.56)	(2.07)
*nCohesion10sq*	−0.256	−0.799 ***	−0.528 **	0.0781	−0.221	−0.246 *
	(−1.38)	(−2.77)	(−2.35)	(0.26)	(−1.59)	(−1.95)
*lnpcgdp*	0.00453 ***	0.00378 **	0.00441 **	0.00387 *	0.00374 ***	0.00595 **
	(2.85)	(2.51)	(2.23)	(1.97)	(2.69)	(2.25)
*fiscal*	0.0102 **	0.0242 ***	0.0158 ***	0.0192 **	0.00280	0.0342 ***
	(2.11)	(3.39)	(2.80)	(2.20)	(0.97)	(3.20)
*lntrade*	−0.000268	0.000201	−0.000215	−0.000402	−0.0000784	−0.000177
	(−0.71)	(0.41)	(−0.49)	(−0.64)	(−0.20)	(−0.37)
*lnays*	−0.0287 ***	−0.0316 ***	−0.0261 **	−0.0442 ***	−0.0373 ***	−0.00638
	(−3.41)	(−2.94)	(−2.42)	(−3.29)	(−5.06)	(−0.54)
*IndStr*	−0.000956	−0.00149	−0.000185	−0.00303 *	−0.00191	−0.00295
	(−0.63)	(−0.93)	(−0.10)	(−1.75)	(−1.30)	(−1.36)
*_cons*	−0.139	−0.584 **	−0.372 **	0.184	−0.0861	−0.188 **
	(−1.05)	(−2.48)	(−2.18)	(0.72)	(−0.76)	(−2.03)
r2_a	0.4248	0.2465	0.2295	0.4271	0.3944	0.3011
F	23.9279 ***	10.0401 ***	19.3293 ***	15.5772 ***	18.0744 ***	.
N	2166	1979	1955	2190	2416	1729

Notes: *t* statistics are in parentheses. *, **, and *** represents *p* < 0.1, *p* < 0.05, and *p* < 0.01, respectively. For concise presentation, the coefficients for the year dummies are excluded from the tables.

**Table 4 ijerph-19-06184-t004:** Mechanism tests: impact of city shape on population growth.

	(1)	(2)
	PopGR1	PopGR2
*nCohesion10*	7.104 **	51.14
	(2.02)	(1.14)
*nCohesion10sq*		−25.10
		(−0.98)
*lnpcgdp*	0.665 **	0.678 **
	(2.14)	(2.16)
*fiscal*	−1.541	−1.596
	(−1.35)	(−1.40)
*lntrade*	0.147 *	0.150 *
	(1.85)	(1.88)
*lnays*	0.469	0.447
	(0.31)	(0.30)
*IndStr*	−0.0780	−0.0754
	(−0.27)	(−0.26)
*_cons*	−7.188	−26.36
	(−1.58)	(−1.30)
r2_a	0.0165	0.0165
F	3.9020 ***	3.7397 ***
N	5348	5348

Notes: *t* statistics are in parentheses. *, **, and *** represents *p* < 0.1, *p* < 0.05, and *p* < 0.01, respectively. For concise presentation, the coefficients for the year dummies are excluded from the tables.

**Table 5 ijerph-19-06184-t005:** Mechanism tests: impact of city shape on housing prices.

	(1)	(2)	(3)	(4)	(5)
	HsPr_real1	HsPr_real2	PHR	sobel1	sobel2
*nCohesion10*	−565.0	−50082.4 **	0.899 ***	−0.0617 ***	0.0177 *
	(−0.32)	(−2.06)	(2.70)	(−4.96)	(1.81)
*nCohesion10sq*		28200.4 **	−0.499 ***		
		(2.02)	(−2.67)		
*HsPr_real*			0.000000125	−0.00000376 ***	−0.00000144 ***
			(0.50)	(−8.17)	(−7.48)
*lnpcgdp*	−341.0 *	−355.7 *	0.00383 ***	0.00146	0.00165 ***
	(−1.67)	(−1.74)	(3.01)	(1.26)	(3.15)
*fiscal*	−3503.5 ***	−3442.1 ***	0.0164 ***	0.0404 ***	0.0197 ***
	(−5.48)	(−5.38)	(3.49)	(5.68)	(6.17)
*lntrade*	13.63	11.37	−0.0000355	−0.00138 ***	−0.00284 ***
	(0.41)	(0.34)	(−0.10)	(−3.63)	(−16.51)
*lnays*	964.1	992.4	−0.0295 ***	0.0586 ***	0.0448 ***
	(0.80)	(0.82)	(−3.68)	(5.79)	(10.55)
*IndStr*	556.4 ***	553.2 ***	−0.00142	−0.00315 **	−0.00295 ***
	(3.35)	(3.36)	(−1.09)	(−1.98)	(−4.29)
*_cons*	64.21	21635.2 **	−0.345 **	−0.0389 *	−0.0820 ***
	(0.02)	(2.06)	(−2.39)	(−1.79)	(−6.63)
r2_a	0.7029	0.7037	0.3275	0.1930	0.2797
F	79.9398 ***	76.9053 ***	20.0068 ***	36.6117 ***	173.0608 ***
N	5364	5364	4145	1043	3102

Notes: *t* statistics are in parentheses. *, **, and *** represents *p* < 0.1, *p* < 0.05, and *p* < 0.01, respectively. For concise presentation, the coefficients for the year dummies are excluded from the tables.

**Table 6 ijerph-19-06184-t006:** Mechanism tests: impact of city shape on transportation.

	(1)	(2)	(3)	(4)	(5)	(6)
	carpc1	carpc2	PHR	bus1	bus2	PHR
*nCohesion10*	0.967	22.99 *	0.439 *	0.114	−3.911 **	0.977 ***
	(0.91)	(1.70)	(1.82)	(0.76)	(−2.06)	(2.95)
*nCohesion10sq*		−12.47	−0.257 *		2.293 **	−0.544 ***
		(−1.59)	(−1.90)		(2.08)	(−2.92)
*lncarpc*			0.00182 **			
			(2.14)			
*bus*						0.00528 **
						(2.03)
*lnpcgdp*	0.439 ***	0.443 ***	0.00172	−0.0391 ***	−0.0403 ***	0.00435 ***
	(6.51)	(6.61)	(1.25)	(−2.80)	(−2.86)	(3.48)
*fiscal*	0.944 ***	0.934 ***	0.00214	−0.335 ***	−0.330 ***	0.0176 ***
	(4.60)	(4.56)	(0.69)	(−6.11)	(−6.07)	(3.72)
*lntrade*	0.0342 *	0.0370 *	0.0000693	0.00160	0.00140	−0.0000217
	(1.72)	(1.91)	(0.20)	(0.49)	(0.44)	(−0.06)
*lnays*	−0.216	−0.219	−0.0268 ***	0.188 **	0.190 **	−0.0307 ***
	(−0.56)	(−0.57)	(−4.29)	(2.52)	(2.54)	(−3.75)
*IndStr*	0.103 **	0.104 **	−0.00245 **	0.0288 **	0.0286 **	−0.00126
	(2.10)	(2.10)	(−2.07)	(2.42)	(2.41)	(−1.02)
*_cons*	1.938	−7.741	−0.129	−0.354	1.398 *	−0.377 ***
	(1.57)	(−1.35)	(−1.19)	(−1.62)	(1.81)	(−2.62)
r2_a	0.9003	0.9004	0.3454	0.2965	0.2984	0.3293
F	384.0011 ***	376.1154 ***	17.2813 ***	9.8630 ***	9.5838 ***	19.7017 ***
N	3848	3848	3509	5308	5308	4106

Notes: *t* statistics are in parentheses. *, **, and *** represents *p* < 0.1, *p* < 0.05, and *p* < 0.01, respectively. For concise presentation, the coefficients for the year dummies are excluded from the tables.

**Table 7 ijerph-19-06184-t007:** Robustness tests: impact of city shape on urban poverty (excluding municipalities).

	(1)	(2)
	Robust1_0	Robust1_1
*nCohesion10*	0.0249	0.876 ***
	(0.78)	(2.64)
*nCohesion10sq*		−0.487 ***
		(−2.61)
*lnpcgdp*	0.00354 ***	0.00382 ***
	(2.77)	(2.99)
*fiscal*	0.0164 ***	0.0157 ***
	(3.52)	(3.40)
*lntrade*	−0.0000641	−0.00000967
	(−0.19)	(−0.03)
*lnays*	−0.0293 ***	−0.0296 ***
	(−3.54)	(−3.59)
*IndStr*	−0.00109	−0.00111
	(−0.83)	(−0.84)
*_cons*	0.0352	−0.335 **
	(1.13)	(−2.32)
r2_a	0.3153	0.3229
F	20.4047 ***	19.6270 ***
N	4069	4069

Notes: *t* statistics are in parentheses. **, and *** represents *p* < 0.05, and *p* < 0.01, respectively. For concise presentation, the coefficients for the year dummies are excluded from the tables. Municipalities here refer to municipalities directly under the central government, which include Beijing, Shanghai, Chongqing, Tianjin.

**Table 8 ijerph-19-06184-t008:** Robustness tests: impact of city shape on location quotient.

	(1)	(2)
	Robust2_0	Robust2_1
*nCohesion10*	3.519	89.93 ***
	(1.41)	(3.07)
*nCohesion10sq*		−49.37 ***
		(−3.03)
*lnpcgdp*	0.246 **	0.275 ***
	(2.48)	(2.82)
*fiscal*	2.470 ***	2.399 ***
	(5.34)	(5.24)
*lntrade*	−0.0400	−0.0347
	(−1.10)	(−0.99)
*lnays*	1.125 *	1.101 *
	(1.74)	(1.74)
*IndStr*	−0.105	−0.106
	(−0.89)	(−0.91)
*_cons*	−4.864 *	−42.43 ***
	(−1.82)	(−3.29)
r2_a	0.1043	0.1218
F	2.7524 ***	3.2425 ***
N	4145	4145

Notes: *t* statistics are in parentheses. *, **, and *** represents *p* < 0.1, *p* < 0.05, and *p* < 0.01, respectively. For concise presentation, the coefficients for the year dummies are excluded from the tables.

**Table 9 ijerph-19-06184-t009:** Robustness tests: other tests.

	(1)	(2)
	Robust3	Robust4
*nCohesion10*	7.270	0.644 **
	(1.06)	(2.22)
*nCohesion10sq*	−7.873	−0.359 **
	(−1.00)	(−2.22)
*nCohesion10power*	2.828	
	(0.94)	
*lnpcgdp*	0.00378 ***	0.00242 **
	(2.98)	(2.03)
*fiscal*	0.0160 ***	0.0200 ***
	(3.45)	(3.92)
*lntrade*	−0.00000320	0.000225
	(−0.01)	(0.69)
*lnays*	−0.0296 ***	−0.0176 **
	(−3.66)	(−2.52)
*IndStr*	−0.00140	−0.00130
	(−1.10)	(−1.16)
*_cons*	−2.168	−0.254 **
	(−1.10)	(−1.99)
r2_a	0.3290	0.4473
F	19.4351 ***	22.3572 ***
N	4145	4145

Notes: *t* statistics are in parentheses. **, and *** represents *p* < 0.05, and *p* < 0.01, respectively. For concise presentation, the coefficients for the year dummies are excluded from the tables.

## Data Availability

Data are available from the authors upon request.

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
