# Peer review of "Does the Compact City Paradigm Help Reduce Poverty? Evidence from China"

_ijerph, 2022, doi:10.3390/ijerph19106184_

Round 1

Reviewer 1 Report

The article addresses an important and interesting issue concerning  the compact cities’ impact on urban poverty. The structure of the paper is appropriately designed. The article is well quite well grounded in the theory. However, I would suggest including recently published articles, e.g.: Bibri, S.E., Krogstie, J. and Kaerrholm, M. (2020), Compact city planning and development: Emerging practices and strategies for achieving the goals of sustainability, Developments in the Built Environment, vol. 4, pp. 1-20; Kain, J.-H., Adelfio, M., Stenberg, J. & Thuvander, L. (2022). Towards a systemic understanding of compact city qualities, Journal of Urban Design, 27(1), pp. 130-147, Liddle, B. (2017), Urbanization and Inequality/Poverty, Urban Science, 1(4), 35, pp. 1-7. Moreover, the Abstract as well as the Introduction should very clearly point out the purpose of the article. I would consider introducing research questions in the latter section, as well. The presenting research methodology should not be limited to several sentences. This section could be developed and adequately described. In my opinion the conclusions could be more supported by results.  

Author Response

Thank you for your careful reading and highly constructive comments. We have greatly benefited from them and have followed your suggestions in revising the whole paper. We provide an attached document to make detailed response to each comment.

Reviewer 2 Report

The authors should address the following comments to improve the quality of the paper:

  • Section 1: the first paragraph should highlight the extent of urbanization globally and in China to buttress the importance of the issue? 
  • Section 1: there is a need in the second paragraph to specify which SDG the compact cities could help achieve. 
  • Section 2:
  • Section 3.2 (page 6): please justify using two different periods (2000-2013 and 2014-2018), and why not more recent data points?
  • Section 3: there is the need to rationalize the choice of all equations and cite some studies that used them.
  • Figures 2 & 3: please increase their sizes and resolutions to be more legible.
  • Section 5: the section is expected to discuss the important discoveries and how they support/corroborate or differ from prior studies and likely explanations. However, not a single study is cited.
  • The discussion should also underscore the value this study adds to the literature and whether the method can be generalized (applicable to other settings).
  • Section 6: What are the key lessons of the study, the implications of the findings for urban policy and practice, as well as its limitations, and future research direction

Author Response

(The authors gave the same response as above.)

Reviewer 3 Report

This study strives to identify the relation of compactness with the level of urban poverty. The authors claim that their empirical tests suggest that the compactness has an inverted U-shape relationship with the incidence of poverty. The strengths of this paper are that it is generally well-written and the study is reproducible as the description of the methods used is detailed and clear. The weaknesses are that there are some claims that need more clarification . As well, the findings warrant a greater discussion. 

1. Abstract should be developed more in order to include the main findings which is the so-called inverted U-shape relation.

2. Please define terms at point of first use, such as "compact cities"  (I was surprised by this; I thought we were all born with innate knowledge of this concept). 

3. Aim to include in the introduction a short discussion on sustainable development (meaning, content) and its relation to poverty. See for instance (i). Manioudis, M.  & Meramveliotakis, G.  (2022) “Broad strokes towards a grand theory in the analysis of sustainable development: a return to the classical political economy”, New Political Economy, DOI: 1080/13563467.2022.2038114. (ii). Cobbinah, P. B., Erdiaw-Kwasie, M. O., & Amoateng, P. (2015). Rethinking sustainable development within the framework of poverty and urbanisation in developing countries. Environmental Development13, 18-32. 

3. The authors must clarify/define in more concrete terms what exactly do they mean by "poor people". This is because poverty is usually associated with the group of extremely poor people. 

4. p.2 , line 1 “Most studies are focused on the impact...”. Please indicate some of these studies.

5. Section 5 must be substantially developed and becomes more robust. For instance, inverted U-shape relation means that at low level of compactness the poverty increases (why? the authors did not clarify) while at some point (which is this point? the authors did not discuss) further enhancement of compactness has the reverse effect to alleviate the poverty. 

6. Section 6 is actually a repetition of section 5.  

Author Response

(The authors gave the same response as above.)

Round 2

Reviewer 3 Report

Happy to see the authors to accommodate the majority of the comments I have made. The quality of the paper has been enriched and the analysis became more robust. However, there are still some points that need more clarification and elaboration:

  1. Aim to include in the introduction a short discussion on sustainable development (meaning, content) and its relation to poverty. See for instance (i). Manioudis, M. & Meramveliotakis, G. (2022) “Broad strokes towards a grand theory in the analysis of sustainable development: a return to the classical political economy”, New Political Economy, DOI: 1080/13563467.2022.2038114. (ii). Cobbinah, P. B., Erdiaw-Kwasie, M. O., & Amoateng, P. (2015). Rethinking sustainable development within the framework of poverty and urbanisation in developing countries. Environmental Development, 13, 18-32

Τhe authors should include the two mentioned references. Thy use the notion of sustainable development. but they never try to provide a (short) definition of the term.

  1. The authors must clarify/define in more concrete terms what exactly do they mean by "poor people". This is because poverty is usually associated with the group of extremely poor people.

Authors’ Reply: Accepted and Explained. Thank you for this reminder. In this article, we are concerned with the extremely poor, i.e. residents whose housing or income is significantly below the local low income standard.

Authors eventually clarify that by “poor people” they mean the “extremely poor” and then provide their own vague definition ( i.e. residents whose housing or income is significantly below the local low income standard) (for instance what exactly do they mean by significantly below?). If authors concerned about extreme poverty, then they have to embrace the World Bank’s definition of the term, people with incomes below US$ 1.9 per day. But then two fundamental analytical problems arise:

  • According to a joint study (2022) – “Four Decades of Poverty Reduction in China: Drivers, Insights for the World, and the Way Ahead” – was undertaken by China’s Ministry of Finance, the Development Research Center (DRC) of the State Council, and the World Bank, with the China Center for International Knowledge on Development (CIKD) acting as the implementing agency “poverty in China is defined only among rural population, reflecting a longstanding view that in China poverty is “fundamentally a rural phenomenon”, as the share of the urban poor tends to be very small (Naughton 2018)” (p. 3) . Hence, extreme poverty in China is found only in rural areas.
  • Throughout the paper the authors use also the notion of low-income households, to denote the extremely poor? (e. g. p. 3) Of course this is not valid. Moreover, if the extreme poor is the case, then some analyses became irrelevant. For instance, section 2.2. The housing impact for the poor may be well- fitted for the low-income households, thought market forces housing prices are reduced then the households are in better position to find a (better) home, but it has no analytical validity for the extreme poor. In the first instance, the housing of the later cannot be merely subject to market forces but to state’s social welfare provisions. The same also holds for section 2.3. The transportation impact for the poor. In general terms, the alleviation of extreme poverty cannot be subject to the market forces analysis as the authors do.

For overcoming all these problems, I suggest that the authors should define “poor people” as the ‘lower-income’ class (lowest bar) covers all individuals with a net income below 50% of median income of the total population (OECD, 2000) minus the extreme poor people with incomes below US$ 1.9 per day.  Doing this the authors establish  a clear definitional demarcation between poverty and extreme poverty, two notions that entail different qualities.

  1. The authors acknowledge that “…we failed to find out the relevant mechanism of why the increase of compactness increases poverty when the compactness is low” and then place this “deficiency” in study’s limitations. I am not sure if this is a legitimate, since it refers to the main finding of the paper (the reverse U-relationship). If this could be characterized as a limitation, is a severe limitation of the study, since you are unable to fully explicate the core finding!

Author Response

1.

Thank you for your valuable advice. We introduced the concept of sustainable development in the introduction and cite two articles. The details are as follows:

“The concept of sustainable development, meaning development that can be sustained in the long term without compromising the ability of future generations to meet their own needs (Manioudis & Meramveliotakis, 2022; Cobbinah et al., 2015). ”

“As urbanisation accelerates in many countries, urban issues will become increasingly important to socio-economic development. As an important aspect of urban issues, it is becoming increasingly important to raise awareness and understanding of poverty in an urban context (Cobbinah et al., 2015; Liddle, 2017). ”

2.

Thank you for your careful and valuable comments. We are sorry that we failed to give a correct explanation in our last reply. Thank you for helping us clarify the concept. Now, we can make it clear that the poor in the paper means the “lower-income” class. We represent this group using the number of people who receive the subsistence allowance.

3.

Thank you for pointing out this deficiency in this paper. We improved on this deficiency to make our core finding better empirically supported. The place where we modify is in the intermediary effect of housing price. In the previous draft, the impact of housing price on urban poverty was positive (Column 3 in Table 5), but the impact here is not significant. We conducted sobel test and found that the mediating effect was significant. Therefore, we believe that the impact of housing price on urban poverty is significantly positive. However, when we re-examine the results of Sobel test, we find that the coefficient estimation result given by Sobel test is significantly negative. Therefore, we believe that the result of Sobel's test should prevail, that is, the impact of housing price on urban poverty is significantly negative. The effect of compactness (via house prices) on urban poverty is then inverted U-shaped. Through the mechanism of housing prices, we can explain very well why, on the left side of the axis of symmetry, compactness increases but poverty rates rise. Because on the left side of the axis of symmetry, increasing compactness leads to lower housing prices, areas with low housing prices become attractive to poor people, increasing the incidence of poverty there.

In addition, we have modified the corresponding part of the conclusion. With regard to the mechanism of influence, both low house prices and good transport play a role in attracting poor people and increasing the incidence of poverty in the local area. However, house prices play a more important role in the process, i.e. the impact of house prices is greater than the impact of public transport on the poor.